# Transition from Natural to Early Synthetic Dyes in the Romanian Traditional Shirts Decoration

**Irina Petroviciu** [1,*], **Iulia Claudia Teodorescu** [2], **Silvana Vasilca** [3,4] **and Florin Albu** [5]

1    National Museum of Romanian History (MNIR), 030026 Bucharest, Romania
2    The ASTRA National Museum Complex, 550399 Sibiu, Romania
3    "Horia Hulubei" National Institute for R&D in Physics and Nuclear Engineering, IRASM Department, 077125 Magurele, Romania
4    Department of Analytical Chemistry, Faculty of Chemistry, University of Bucharest, Panduri 90-92, 050663 Bucharest, Romania
5    Agilrom Scientific SRL, 010193 București, Romania
*    Correspondence: irinapetroviciu@mnir.ro

**Abstract:** The traditional shirt ("*ie*") is the most well-known element of Romanian anonymous textile art. Apart from aesthetic and utilitarian roles, it has strong symbolic significance, mainly through the colours used for decoration. Very recently, the traditional shirt with decoration over the shoulder ("*ia cu altiță*") was introduced as a Romanian identity element as part of UNESCO heritage. Depending on the ethnographic area, the traditional shirt with decoration over the shoulder has acquired special expressive particularities over time. Particularly relevant is that from Valea Hârtibaciului, an area of Transylvania in the very centre of Romania. Although sober in appearance with large fields of white plain weave, it is discreetly decorated with elaborated embroidery on the sleeve bracelets, over the shoulders and neck. Even the colour range and decoration motifs remain unchanged in time, evolution in the materials used and a subtle transition from natural hues to more strident alternatives were observed in the late 19th and early 20th centuries. For the present study, samples were taken from representative objects in the collections of the ASTRA Museum, Sibiu and Ethnographical Museum, Brasov, documented as belonging to the area of Valea Hârtibaciului and dated in the museum archives as from the late 19th and early 20th century. The textile materials and the dyes used in the shirts' embroidery were monitored. Fibre identification was made by optical microscopy and infrared spectroscopy (FTIR-ATR). Dye analysis was performed by liquid chromatography coupled with UV-Vis (diode array) detection, while some of the samples were also analysed by liquid chromatography coupled with mass spectrometric detection (LC-DAD-MS). Dyes were extracted from the fibres by acid hydrolysis. Identification was based on data collected on standards, dyes and dyed fibres. For the early synthetic dyes, a dedicated library of references was built, which includes information relative to the most relevant representatives used between 1850 and 1900, the 'Helmut Schweppe list'. According to the study, in the last decades of the 19th century, natural dye sources such as dyer's broom, madder, Mexican cochineal and indigoid dyes were gradually replaced by early synthetic dyes: fuchsine (1856), methyl violet (1861), synthetic alizarin (1871), brilliant green (1879), azo flavine 3R (1880), rhodamine B (1887) and others.

**Keywords:** natural dyes; early synthetic dyes; textiles; liquid chromatography; identification; shirts; Romania

## 1. Introduction

The shirt is the most significant part of the Romanian traditional costume and therefore receives particular attention, reflected in cut, decoration, chromatics, materials and techniques [1]. On special occasions, it acquires even a stronger symbolic significance, becoming a vestmental code to express provenience, appurtenance to age and ethnic groups, or civil status [1,2]. At the end of the 19th century, this traditional shirt elegance attracted the

attention of the Romanian Queen Elizabeth (1843–1916), who started to wear shirts at Court events and international meetings. Her aim was to evidence that Romanian spirituality has deep roots in the history of the nation and that it is also witnessed the refinement and artistic diversity of traditional clothing. Noblewomen followed the Queen's model in her effort to promote the traditional costume, while later on Queen Mary of Romania (1875–1938) and her daughters also became ambassadors of the traditional costume [3]. Nowadays, traditional shirts dating back to the 19th and 20th centuries are preserved in museum collections, as material testimonies of the peasants' skillfulness and artistic taste. An increasing interest in their examination has materialized in recent years in several studies and exhibitions [1,2]. Moreover, the traditional shirt with decoration over the shoulder, the so called "*ia cu altiță*"—still widespread nowadays on both sides of the Carpathian mountains—was recently accepted as part of the UNESCO National Inventory of Active Intangible Cultural Heritage Elements [2]. Since 2012, the concern to conserve the traditional shirt as an element of cultural identity was also reflected in the spontaneous organisation of several sewing groups, in different areas of Romania, who produce their own shirts by following models of traditional items. In order to reproduce them as faithfully as possible, special attention is given to the materials—textile fabrics, sewing and decoration yarns, as well as to the dyes used.

Natural dyes were the only source of colour for textile dyeing since antiquity and until synthetic alternatives became available. The first successful efforts to develop new colouring compounds date from 1740 to 1771, when indigo carmine and picric acid were first synthesized [4]. Nevertheless, the triumph of synthetic dyes started in 1856, when the first synthetic dye, mauveine, was accidentally discovered [4,5]. In the following years, a large number of synthetic dyes with different structures came on the market, about 400 being in use before 1900, which resulted in the rapid decline of the natural dyes [4]. Regarding the period between 1850 and 1900, the literature mentions that in the three Romanian provinces (Wallachia, Moldavia and Transylvania) a large number of local plants were employed for dyeing purposes [6–9]. An invaluable treasure for documentation of rural life at the end of the 19th century is a collection of dyeing recipes edited by the Romanian Academy in 1914, which includes about 300 descriptions. Beside the large number of natural local sources discussed, the book makes reference to imported natural dyes and early synthetic alternatives [10].

The use of natural and synthetic dyes in historic textiles is nowadays revealed by analytical investigation. Since its first use in 1985 [11], liquid chromatography with UV-Vis (diode-array) detection became the standard method for dye identification and characterisation [12–17], while more recently mass spectrometers were added to the configuration due to their increased sensitivity, which results in lower detection limits, more accurate identifications and smaller sample consuming [18–25]. Report describing historic textiles research based on natural dye detection remain of major interest; however, an increasing number of publications refer to the early synthetic dyes' characterization and identification [4,5,26–32].

Since 1997, a project aiming to enrich the existing information on textiles in the Romanian collection based on the analytical investigation of dyes has been developed. A particular attention was given to traditional textiles through studies on representative items from several museum collections in Romania (see Acknowledgements) [33–38]. Research brought analytical evidence that natural local and imported dyes and synthetic colours co-existed in villages in the three Romanian provinces in the 19th and 20th century. A collection of shirts dated between 1850 and 1930 are catalogued within the cultural project *Saving the others' culture*. Research on home-made shirts with decoration over the shoulder, from Valea Hârtibaciului [39] seemed the ideal opportunity to better understand the transition from natural to early synthetic alternatives. Although sober in appearance due to the large fields of white plain weave, the shirts from Valea Hârtibaciului (an area of Transylvania, in the very center of Romania) are discreetly decorated on the sleeve bracelets, over the shoulders and the neck, with elaborated embroidery (Figure 1). The case study selection was based on the observation that, depending on the ethnographic area, the shirt with decoration

over the shoulder has acquired special expressive particularities over time, which is also reflected by the textile materials and dyes used.

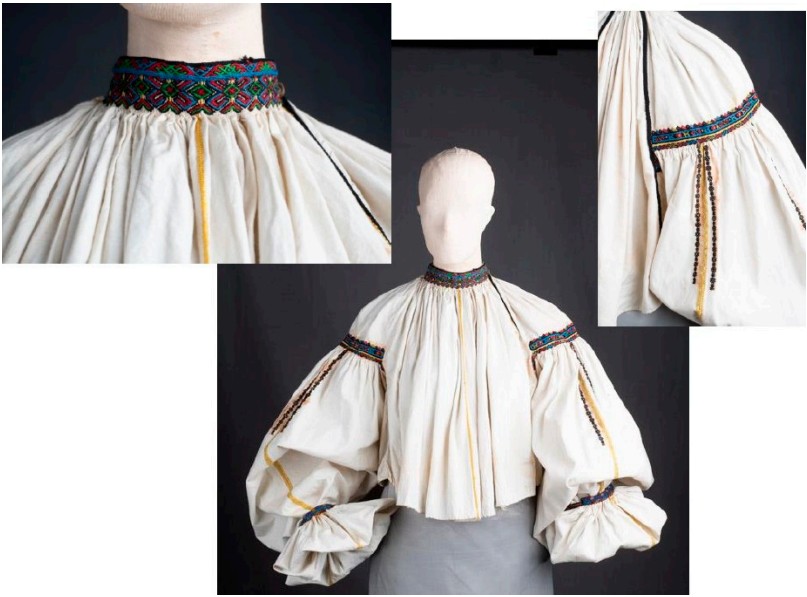

**Figure 1.** Image of shirt inv. 3816 (Muzeul ASTRA Sibiu) to illustrate the overall look of a traditional shirt from Valea Hârtibaciului and details of neck and shoulder decorations.

This article analytically documents the use of natural and early synthetic dyes in hand-made shirts with decoration over the shoulder from Valea Hârtibaciului. A total of 50 samples from 12 shirts belonging to collections of the Ethnographical Museum, Brasov, and the ASTRA Museum, Sibiu, were available for study. One more shirt which was part of a previous survey [35] but typologically belonging to the same group was also included in the discussion.

## 2. Materials & Methods

### 2.1. Sample Description, Documentation and Pretreatment

Samples about 0.5–1 cm long were taken from the back of the shirts' decoration during the conservation procedures. Fibers were first observed under the optical microscope at 10–80× magnification and yarns about 0.5 cm (~3 mg) were cut from the original samples and non-destructively analyzed by attenuated total reflectance infrared spectroscopy (FTIR-ATR). Some of the samples were also analyzed by X-ray fluorescence spectrometry (XRF) for mordants or inorganic colorant identification.

Dyes were extracted from the yarns by the standard acid hydrolysis method [11]. This was preferred to the mild extraction alternatives due to a database of natural dyes which was already in use, and contains information on dyes extracted from standard dyed yarns (yarns dyed in the laboratory with well-documented biological sources) by acid hydrolysis. This was the decision assumed and its limits, caused by the decomposition of glycosides to their parent aglycons, in the case of (natural) flavonoid dyes, were known [40–43]. To each yarn, 200 μL mixture 37% $HCl/CH_3OH/H_2O$ 2:1:1 ($v/v/v$) were added and the mixtures were kept at 100 °C for 10 min. The solutions were evaporated to dryness in a vacuum desiccator. Each sample was redissolved in 100 μL solution $CH_3OH/H_2O$ 1:1 ($v/v$) and centrifuged at 12,000 rpm for 10 min. The supernatants were transferred into chromatographical vials and injected into the chromatographical system. When the presence of indigo based dyes was suspected, as in the cases of visual blue, green, violet and black samples, a second extraction in 100 μL dimethyl sulfoxide (DMSO) was made, with the samples kept at 80 °C for 10 min. The two solutions were merged and thus analyzed.

*2.2. Fiber Identification*

Fiber documentation and image collection were made with a Nikon SMZ 1000 stereo-microscope coupled with a Nikon DSLR camera, model D3100 Kit AF-s 18–55 mm VR DX. Further fiber investigation was made by infrared spectroscopy (FTIR-ATR), where a Bruker Optics Alpha spectrometer equipped with a Platinum ATR single reflection diamond ATR module was used. Spectra were acquired in the 4000–400 cm$^{-1}$ domain, with a resolution of 4 cm$^{-1}$. Spectra collection and data processing were made with dedicated software, Opus 7.0.

*2.3. Dye Analysis by Liquid Chromatography, Instrumentation and Parameters*

All the samples were analyzed by liquid chromatography with UV-Vis (diode-array) detection, LC-DAD on System 1, while selected samples were also investigated by liquid chromatography with UV-Vis and mass spectrometric detection, LC-DAD-MS on System 2.

2.3.1. System 1 (LC-DAD)

An Agilent 1260 Infinity II series liquid chromatograph (Agilent Technology, Santa Clara, CA, USA) consisting of a quaternary pump (G7129A), a standard autosampler (G7111B), a column thermostat (G7116A) and multi-channel diode-array detector (G7115A) was used for dye analysis. OpenLAB CDS software was used for the chromatographic system control, data acquisition and processing. A Zorbax C18 column, 150 mm length, 4.6 mm i.d. and 5 μm particle size, was thermostated at 40 °C. The mobile phase consists of a mixture of aqueous 0.2% (*v/v*) formic acid (solvent A) and methanol/acetonitrile (1:1, *v/v*) plus 0.2% formic acid (as solvent B). Gradient elution was applied, by using the following profile: at 0 min, 15% solvent B; from min 0 to 5, linear increase to 25% solvent B; from min 5 to 10, constant at 55% solvent B; from min 10 to 16, linear increase to 100% solvent B; from min 16 to 18, constant at 100% solvent B; and step jump at 15% solvent B, with 5 min re-equilibration period between runs (post-time). The flow rate was set at 0.8 mL/min and the injected volume of the sample was 10 μL. The UV–Vis spectra were acquired in the range from 200 to 900 nm, with a simultaneous monitoring at five wavelengths (255, 275, 295, 420 and 490 nm), having a frequency of 0.03 min and a resolution of 2 nm.

2.3.2. System 2 (LC-DAD-MS)

An Agilent 1260 LC system was used, composed of the following modules: quaternary pump (Model G1311C), automatic injector (G1367E) and column thermostat (G1316C). The diode array detector (G4212A) and the triple quadrupole mass spectrometer (G6410B) were serially connected. The latter used an ESI ionization source (ESI, Model G1948B), operated under negative and positive ion monitoring modes. Chromatographic separation was the same as for System 1, except for Solvent B which was methanol/acetonitrile 1:1 (*v/v*) with no formic acid content. No extra samples were prepared for LC-DAD-MS analysis, 10 μL from the samples already prepared for LC-DAD being injected in the chromatographic system. UV–Vis spectra were acquired with a DAD detector which was placed between the column and the MS ion source. Spectra were collected over the 190–640 nm range, with a resolution of 2 nm. For the MS detector, the following ESI operation parameters were used: drying gas temperature 350 °C; drying gas flow 8 L/min; pressure of the nebulising gas 40 psi; Vcap 2500 (−) in negative ion mode and 2500 (+) in positive ion mode. The triple quadrupole used MS2 type scan when used as a single MS instrument; the data storage was set on profile and the peak width at 0.07; fragmentor 135 V; ΔEMV 400 V. The scanning interval for the mass to charge ratio (*m/z*) was between 100 and 600 a.m.u., and acceleration voltage on the collision cell: 7 V; Dwell Time 500 ms. In order to control the chromatographic system, as well as for data acquisition and processing, Agilent MassHunter Quantitative Analysis B.06.00 software was used. The analytical procedure was described in detail in an earlier publication, where an ion trap mass spectrometer was used instead of the triple quadrupole [44]. The selected samples were first analyzed

with single MS detection in the Full Scan mode and the resulting data was processed by extracting chromatograms, according to the molecular ions of the dyes in the database.

### 2.4. Dyes Attribution, Databases

Dyes were attributed based on their retention and UV-Vis data, according to information collected on standards, dyes and dyed yarns. MS data were also considered for identification when the samples were investigated by LC-DAD-MS. As described in previous publications [35], biological source attribution was based on data collected on yarns dyed in the laboratory, by following traditional dyeing recipes. The existing database which mainly contained natural dyes was enriched with information on synthetic products, in order to correspond to actual needs. Experiments were thus performed on dyes and dyed yarns (prepared in the laboratory with contemporary products acquired from various providers), as well as on the Schweppe collection of early synthetic dyed yarns. The latter is a collection of yarns dyed with a selection of 65 synthetic dyes, considered by Helmut Schweppe as the most commonly used between 1850 and 1900 [26,45]. It was offered by the chemist to European and worldwide museums in the 1960–1970, during a series of dyeing workshops with these well-documented products. The collection used in the present study was prepared by Ronnee Barnett, textile restorer at the Metropolitan Museum of New York (MET) and offered by Florica Zaharia, Muzeul Textilelor Băița Romania (see Section Acknowledgements). Analytical information from the in-house built database was supplemented with literature data on early synthetic dyes [4,5,26–29]. Retention and UV–Vis data on the natural dyes and biological sources discussed is given in Table 1, while similar information on the early synthetic alternatives can be found in Table 2. When mass spectrometric data was used for dye attribution (samples analyzed with System 2), it was added in a separate column in each of the two tables.

### 2.5. X-ray Fluorescence Spectrometry (XRF)

Elemental analysis of selected samples was performed with a portable XRF spectrometer Bruker S1 TITAN Model 600, with the following specifications: rhodium (Rh) tube, silicon drift chamber detector (SDD), 5 mm spot size. The system used was air-path, elemental range $Z > 12$ (Mg). It was set to 45 kV/10 µA for heavy elements and to 15 kV/27 µA for the easy.

**Table 1.** The biological sources discussed, the corresponding dyes and their retention and UV–Vis data. MS data for the dyes identified by LC-DAD-MS were also included.

| Biological Source Common and Latin Name(s) | Dye | Abbreviation | Retention (min.) | | UV_Vis Data | MS Data |
|---|---|---|---|---|---|---|
| Madder (Mad) (*Rubia tinctorum* L.) | Alizarin | al | 15.4 | | 202; 248; 278; 430 | - |
| | Purpurin | pu | 16.4 | | 204; 256; 294; 480 | - |
| Indigo based (see text) (Ind) Woad (*Isatis tinctoria*) or Indigo (*Indigofera* sp.) | Indigotin | ind | 16.2 | | 238; 285; 330; 610 | - |
| Dyer's broom (DB) (*Genista tinctoria* L.) | Luteolin | lu | 13.0 | | 208; 254; 266; 348 | - |
| | Genistein | ge | 13.7 | | 208; 260 | - |
| | Apigenin | ap | 13.9 | | 210; 268; 336 | - |
| Sawwort (Saw) (*Serratula tinctoria* L.) | Luteolin | lu | 13.0 | 12.7 | 208; 254; 266; 348 | MS− 285 |
| | Apigenin | ap | 13.9 | 13.7 | 210; 268; 336 | MS− 269 |
| | 3-O-methyl-quercetin | 3-O-methyl-qu | - | 12.9 | - | MS− 315 |
| Carminic acid based Mexican cochineal (*Dactylopius coccus* C.) (ca) (see text) | Carminic acid | ca | 8.8 | 9.0 | 226; 276; 310; 494 | MS− 491 |
| Tannins (tan) | Ellagic acid | ea | 9.9 | 10.0 | 254; 366 | MS− 301 |

**Table 2.** Early synthetic dyes detected, their retention and UV–Vis data. MS data for the dyes identified by LC-DAD-MS were also included.

| Dye | Color Index | Retention (min.) | | UV-Vis Data | MS Data |
|---|---|---|---|---|---|
| | | System 1 (LC-DAD) | System 2 (LC-DAD-MS) | | |
| Alizarin (synthetic) (1871) (Al) anthrapurpurin, flavopurpurin, alizarin | Mordant Red II CI 58,000 | 13.6 | - | 274; 338; 428 | - |
| | | 13.8 | - | 272; 406 | - |
| | | 15.2 | - | 248; 278; 430 | - |
| Alkali blue (1864) | Acid Blue 110 CI 42,750 | 14.4 | - | 596 | - |
| | | 16.6 | - | 596 | - |
| Auramine (1883) (Aur) | Basic Yellow 2 CI 41,000 | 14.2 | - | 360 | - |
| | | 15.7 | - | 366; 486 | - |
| | | 16.9 | - | 368 | |
| Azo Flavine 3R (1880) (AzoFl) | Acid Orange 1 CI 13090 | 16.2 | 15.9 | 410 | MS− 258 |
| Brilliant green (1879) (Bri Gr) | Basic Green 1 CI 42,040 | 15.4 | 15.9 | 428; 626 | MS+ 385 |
| Fuchsine (magenta) (1856) (Fuch) pararosaniline, fuchsine (rosaniline), methylrosaniline, dimethylrosaniline | Basic Violet 14 CI 42510 | 10.6 | | 544 | - |
| | | 11.2 | | 546 | - |
| | | 11.9 | | 548 | MS+ 316 |
| | | 12.5 | | 550 | MS+ 330 |
| Methyl Violet (1861) (Me-Vio) | Basic Violet 1 CI 42,535 | 13.9 | | 568 | - |
| | | 14.2 | 14.5 | 574 | MS+ 344 |
| | | 14.6 | 15 | 582 | MS+ 358 |
| | | 15.0 | 15.4 | 590 | MS+ 372 |
| Cristal Violet (1883) | Basic Violet 3 CI 42,555 | 15.0 | 15.4 | 590 | MS+ 372 |
| Orange GG (1878) (Ora GG) | Acid Orange 10 CI 16,230 | 12.2 | | 480 | - |
| Picric acid (1771) (Pic Ac) | Acid Dye CI 10,305 | 12.5 | | 360 | - |
| Safranin T (1859) (Saf T) | Basic Red 2 CI 50,240 | 11.7 | | 530 | - |
| | | 12.1 | | 526 | - |
| Rhodamine B (1887) (Rhod) | Basic Violet 10 CI 45,170 | 15.1 | 15.2 | 557 | MS+ 443 |
| Uranine A (1871) (Uran) | Acid Yellow 73 CI 45,350 (20) | 14.0 | | 482 | - |
| | | | | | - |
| Victoria Blue B (1883) (Vic B) | Basic Blue 26 CI 44,045 | - | 15.3 | - | MS+ 442 |
| | | 15.4 | 15.8 | 306; 346; 604 | MS+ 456 |
| | | 15.7 | 16.1 | 308; 350; 616 | MS+ 470 |

## 3. Results

Results are presented in Table 3 and will be discussed in the following sections according to the visual colours of the yarns examined and the dyes identified.

**Table 3.** Objects, samples and the results obtained by dye analysis. See Table 1 for the natural dyes abbreviations.

| No. | Sample Code Colour (Fibre) | Sample Location | Dye | Dye Source (Common Name) |
|---|---|---|---|---|
| | | Shirt, inv. 1252, Ethnographical Museum, Brasov (end 19th century) | | |
| 1 | 1252_P1 Red (wool) | Shoulder decoration, left | al, pu | Madder |
| 2 | 1252_P2 Orange (cotton) | Sleeve bracelet, right | - | Lead chromate (PbCrO4) (Pb and Cr confirmed by XRF) |
| 3 | 1252_P3 Blue (wool) | Sleeve bracelet, right | ind | Indigo based |
| 4 | 1252_P4 Green (wool) | Sleeve | lu, ge, ap | Dyer's broom (with copper mordant) (Cu confirmed by XRF) |
| | | Shirt, inv. 1415, Ethnographical Museum, Brasov (end 19th century) | | |
| 5 | 1415_P1 Red (wool) | Neck decoration | al, pu | Madder |
| 6 | 1415_P2 Orange (wool) Analizat MS | Neck decoration | 9.2 min (max 273, 482) MS− 330 The dye do not match any of the dyes in the database | |
| 7 | 1415_P3 Blue (wool) | Neck decoration | ind | Indigo based |
| 8 | 1415_P4 Green (wool) | Neck decoration | lu, ge, ap, ind | Dyer's broom + Indigo based |
| | | Shirt with lines over the elbow, inv. 1422, Ethnographical Museum, Brasov (end 19th–early 20th century) | | |
| 10 | 1422_P1 Red (wool) | Sleeve bracelet, left | al, pu, ea | Madder and tannins |
| 11 | 1422_P2 Red (cotton) | Sleeve bracelet, left | anthrapurpurin flavopurpurin al | Synthetic alizarin (1871) |
| 12 | 1422_P3 Red-pink (silk) | Sleeve bracelet, left | Rhodamine B (1887) | |
| 13 | 1422_P4 Yellow (wool) | Sleeve bracelet, left | lu, ge, ap | Dyer's broom |
| 14 | 1422_P5 Blue (wool) | Sleeve bracelet, right | ind | Indigo based |
| | | Shirt, inv. 1429, Ethnographical Museum, Brasov (early 20th century) | | |
| 15 | 1429_P1 Red (silk) | Neck decoration | anthrapurpurin flavopurpurin al ca | Synthetic alizarin (1871) Carminic acid based |
| 16 | 1429_P2 Violet (silk) | Neck decoration | Fuchsine (magenta) (1856) | |
| 17 | 1429_P3 Violet (wool) | Neck decoration | Rhodamine B (1887) | |
| 18 | 1429_P4 Yellow (silk) | Neck decoration | Azo Flavine 3R (1880) | |
| 19 | 1429_P5 Blue (silk) | Shoulder decoration | Victoria Blue B (1883) | |

**Table 3.** *Cont.*

| No. | Sample Code Colour (Fibre) | Sample Location | Dye | Dye Source (Common Name) |
|---|---|---|---|---|
| | | Shirt, inv. 6847, Ethnographical Museum, Brasov (second half 19th century) | | |
| 20 | 6847_P1 Orange (cotton) | Shoulder decoration, right | - | Lead chromate (PbCrO4) (Pb and Cr confirmed by XRF) |
| 21 | 6847_P2 Red (wool) | Shoulder decoration, right | al, pu | Madder |
| 22 | 6847_P3 Blue (wool) | Shoulder decoration | ind | Indigo based |
| 23 | 6847_P4 Yellow (cotton) | Shoulder decoration | - | Lead chromate (PbCrO4) (Pb and Cr confirmed by XRF) |
| | | Shirt, inv. 194P, ASTRA Museum, Sibiu (second half 19th century) | | |
| 24 | 194_P1 Red (wool) | Neck decoration | al, pu | Madder |
| 25 | 194_P2 Green (wool) | Shoulder decoration, right | lu, ge, ap, ind | Dyer's broom Indigo based |
| 26 | 194_P5 Violet (silk) | Shoulder decoration | ca MS− 491 | Carminic acid based |
| 27 | 194_P6 Brown (silk) | Shoulder decoration | ea MS− 301 | Tannins |
| 28 | 194_P7 Blue (wool) | Shoulder decoration | ind | Indigo based |
| | | Shirt, inv. 177P, ASTRA Museum, Sibiu (end 19th–early 20th century) | | |
| 29 | 177_P1 Pink (silk) | Shoulder decoration | | Rhodamine B (1887) |
| 30 | 177_P2 Violet (wool) | Shoulder decoration | | Methyl Violet (1861) |
| | | Shirt, inv. 169P, ASTRA Museum, Sibiu (second half 19th century) | | |
| 31 | 169_P1 Pink (silk) | Shoulder decoration | ca | Carminic acid based (with tin mordant) (Sn confirmed by XRF) |
| 32 | 169_P2 Blue (pale) (silk) | Shoulder decoration, right | ind | Indigo based |
| 33 | 169_P3 Blue (dark) (silk) | Neck decoration | ind | Indigo based |
| 34 | 169_P4 Brown (silk) | Shoulder decoration, right | ea MS− 301 | Tannin based with iron mordant (Fe confirmed by XRF) |
| | | Shirt, inv. 3728, ASTRA Museum, Sibiu (last decade 19th–early 20th century) | | |
| 35 | 3728_P1 Red (silk) | Shoulder decoration | | Orange GG (1878) and Methyl Violet (1861) (contamination, see coments in the text) |
| 36 | 3728_P2 Green (silk) | Shoulder decoration | | Brilliant Green (1879) |
| 37 | 3728_P3 Violet (silk) | Shoulder decoration | | Fuchsin (1856) and Methyl Violet (1861) |
| 38 | 3728_P4 Blue (silk) | Shoulder decoration | | Victoria Blue B (1883) |

**Table 3.** *Cont.*

| No. | Sample Code Colour (Fibre) | Sample Location | Dye | Dye Source (Common Name) |
|---|---|---|---|---|
| | | Shirt, inv. 3816, ASTRA Museum, Sibiu 1885 | | |
| 39 | 3816_P1 Red (silk) | Shoulder decoration | ca | Carminic acid based |
| 40 | 3816_P2 Green (silk) | Shoulder decoration | | Picric acid (1771) |
| 41 | 3816_P3 Blue (silk) | Shoulder decoration | | Alkali Blue (1864) |
| 42 | 3816_P5 Green (wool) | Neck decoration | | Brilliant Green (1879) |
| 43 | 3816_P6 Blue (lână) | Neck decoration | | Victoria Blue B (1883) |
| 44 | 3816_P7 Yellow (silk) | Shoulder decoration | | Auramine (1883) |
| | | Shirt, inv. 6464, ASTRA Museum, Sibiu (last decade 19th–early 20th century) | | |
| 45 | 6464_P1 Red (silk) | Shoulder decoration | | Rhodamine B (1887) |
| 46 | 6464_P2 Pink (silk) | Shoulder decoration | | Rhodamine B (1887) |
| 47 | 6464_P3 Blue (silk) | Shoulder decoration | | Victoria Blue B (1883) |
| 48 | 6464_P4 Yellow (silk) | Shoulder decoration | | Uranine A (1871) |
| | | Shirt, inv. 210, ASTRA Museum, Sibiu (second half 19th century) | | |
| 49 | 210_P1 Pink-violet (silk) | Shoulder decoration | ca | Carminic acid based |
| 50 | 210_P2 Violet (silk) | Neck decoration | | Methyl Violet (1861) |
| | | Shirt, inv. 20P, ASTRA Museum, Sibiu (end 19th century) (samples analysed by LC-DAD-MS [35] | | |
| 51 | 20_P1 Pink (silk) | Neck decoration | ca, fk, ka | Carminic acid based |
| 52 | 20_P2 Pink (pale) (silk) | Neck decoration | ca | Carminic acid based |
| 53 | 20_P3 Black (silk) | Shoulder decoration | ca | Carminic acid based and Prussian Blue (Pr Blue) * (Fe confirmed by XRF) |
| 54 | 20_P4 Blue-green (silk) | Shoulder decoration | ind, lu | Indigo based Luteolin based |
| 55 | 20_P5 Blue (silk) | Neck decoration | ind | Indigo based |
| 56 | 20_P6 Brown (wool) | Neck decoration | - | Natural dyed wool |
| 57 | 20_P7 Ochre yellow (silk) | Neck decoration | lu, ge, ap, 3-O-methyl qu | Dyer's broom and sawwort |

* Identified by FTIR/ATR b (signal at 2074 cm$^{-1}$) and confirmed by the presence of Fe in XRF.

### 3.1. Red and Violet

21 samples from the 50 studied have a visual red or violet colour. About half of these samples (9/21) turned out to be naturally dyed while in one other case a combination of natural and synthetic dyes was used. Madder and carminic acid based dyes were the only natural sources of red and violet. Madder was responsible for the colour in half of the red samples (5/10), in all cases being used on wool. Carminic acid based dyes were revealed as single dye source in one red and four violet (or pink–violet) hues, always on silk. Carminic acid was also detected in two more pink silk samples in the shirt previously studied as well as in another red silk sample from the present group, in a dyeing combination with synthetic alizarin.

Identification of madder (*Rubia tinctorum* L.) was based on the presence of the two main anthraquinones in the plant roots, alizarin and purpurin, attributed according to retention and UV-Vis data. The absence of the other anthraquinones in the roots, munjistin, anthragallol, xantho-purpurin and rubiadin, could be explained by their low amount in the acid hydrolyzed samples, under the detection limit of the UV-Vis detector. This observation is supported by previous studies performed in the same research group where the above-mentioned anthraquinones were identified in samples prepared in the same conditions but investigated by using a sensitive mass spectrometer detector [46]. Identification of madder in traditional Romanian textiles is in perfect correlation with the literature, as a Romanian plant encyclopedia dated 1906 refers to madder as local spontaneous plant (*roibă*, *brociu* etc.) sometimes also cultivated for dyeing [6], and a study on Transylvanian traditional textiles with a remark to technology and aesthetics also mentions madder as main source of red [9].

Carminic acid is the main dye component in various insects, with *Porphyrophora species* and *Dactylopius coccus* as the most well-known representatives. Identification of species in historical textiles dyeing is based on the relative calculation regarding carminic acid and minor components, such as dcII (flavokermesic acid C glycoside), kermesic and flavokermesic acids. However, in the present study such interpretation is not possible, due to the reasons explained above when the absence of the minor components in madder roots was discussed. It is very probable that *Dactylopius coccus* (Mexican Cochineal) was used, according to previous studies on traditional Romanian textiles and correlating with the literature [10,33–35]. This supposition is also supported by the use of tin chloride as mordant, as suggested by the presence of tin according to elemental analysis in one of the samples where carminic acid was detected (169_P1).

Synthetic dyes were detected in 11/21 samples, including one where a synthetic dye was detected in a dyeing combination with a natural source (synthetic alizarin and carminic acid based). Apart from synthetic alizarin mentioned above, three other synthetic dyes were detected: fuchsine (in two cases, once together with methylviolet), methyl violet (four, including the one mentioned before) and rhodamine B (five cases). Fuchsine (magenta) (1856) and methyl violet (1861) are representatives of the first class of synthetic dyes, triarylmethane, usually composed by several chromophores which could be evidenced based on the UV-Vis data or the molecular cations (M+) when analyzed in positive ESI mode. In the present study, fuchsine was identified as the unique dye source in one violet silk sample (analized by LC-DAD) and in a dyeing combination with methyl violet, in a red silk one (additionally examined by LC-DAD-MS). For the former (1429_P2), the four dyes described in the literature as pararosaniline (λmax = 544 nm), rosaniline (λmax = 546 nm), methylrosaniline (λmax = 548 nm) and dimethylrosaniline (λmax = 550 nm) [5,26,27] were present while, in the latter (3728_P3), only methylrosaniline (λmax = 548 nm, MS+ 316) and dimethylrosaniline (λmax = 550 nm, MS+ 330) were detected. According to the literature [27], this could be explained by the differences in the synthesis, two production processes being reported, which would result in the so called "early fuchsine" and "late fuchsine", respectively. "Early fuchsine" is characterized by a mixture of the four compounds mentioned above and was obtained in the late 19th century by heating an oxidant with a coal tar distillate containing a mixture of aniline and toluidine in various ratios, either with or without further inclusion of carbon tetrachloride. "Late fuchsine" would be the result of a later process that involved reaction of 4,4′methylene-di-o-toluidine and o-toluidine

with an oxidant, to give mainly methylrosaniline (also called magenta II) and dimethylrosaniline (new fuchsine). Consequently, sample 1429_P2 is more likely to have been dyed with "early fuchsine" and 3728_P3 with "late fuchsine" and methyl violet. Both versions of fuchsine were previously identified in other traditional textiles in Romanian collections, mainly on wool but also on silk [33,37]. The other representative of the triarylmethane group, methyl violet, was detected based on the presence of the three chromophores: λmax = 568 nm, λmax = 574 nm (MS+ 344), λmax = 582 nm (MS+ 358) and crystal violet λmax = 590 nm (MS+ 372). It was evidenced in three samples: a wool violet, a silk violet (in a dyeing combination with fuchsine) and a red silk (in a dyeing combination with Orange GG 3728_P1). However, the putting together of an acid dye (Orange GG) and a basic one (methyl violet) is unlikely, which means that the latter, which was present in another sample from the same shirt, should be considered as contamination. Both methyl violet and fuchsine were also detected in Saxon sheepskin coats' silk decoration, dated between 1892 and 1908, examined in a previous study [38].

Identification of synthetic alizarin was based on the detection of alizarin and the two marker compounds, anthrapurpurin (λmax = 430 nm) and flavopurpurin (λmax = 408 nm). Synthetic alizarin was introduced in 1868, when Graebe and Liebermann made the first artificial synthesis of a natural organic dye, and became available in 1871 [47,48]. It was one of the first synthetic dyes to be used extensively in Romanian traditional textiles, which may be explained by its qualities, as it was considered exceptionally fast against light and washing [47]. It was frequently detected, mainly on cotton, either as an individual dye to achieve a red hue or together with chrome yellow for orange [37]. Its identification, in the present study, in a dyeing combination with carminic acid based dye in a red silk yarn (1429_P1) is illustrative of the experiments performed in the synthetic dyes' debut period.

Rhodamine B (1887), which belongs to the class of xanthene, is detected based on information available in the literature and confirmed by examination of the standard dyed wool samples, λmax = 558 nm and MS+ 443 (Figure 2). These characteristic features were observed in one violet wool and two pink silk samples, as individual dye, and in a red silk sample together with a red dye, which did not match any of the dyes in the existing database but presents UV-Vis data related to safranin T. Unfortunately, no mass spectrometric data was available for the respective sample. Rhodamine B was also identified in Saxon sheepskin coats' silk decoration, dated between 1900 and 1908, in a previous study [38].

### 3.2. Yellow and Orange

Yellow dyes were detected in seven samples described as yellow and green. Dyer's broom, a natural dye, was responsible for the colour in all the samples where wool was implied (4/7), while synthetic dyes were used on silk (3/7). The presence of dyer's broom was suggested by the detection of luteolin (λmax = 348 nm), genistein (λmax = 260 nm) and apigenin (λmax = 336 nm), as compared with data collected on standards. Where a green hue was intended, either copper sulphate as mordant (1252_P4) or indigo based dyes were used (1415_P4 and 194_P2). According to previous studies [33,34,49], dyer's broom was the most commonly preferred biological source in Romanian traditional textiles, which is also supported by its frequent mention in the collection of dyeing recipes edited by the Romanian Academy in 1914, cited above (see Introduction section) [10]. Uranine A (1871) from the class of xanthenes, azo flavine 3R (1880) and the diphenilmethane auramine O (1883) are the three synthetic dyes identified based on the UV-Vis data (λmax = 482 nm for uranine A, λmax = 410 nm for azo flavine 3R and λmax = 366, 486 nm for auramine O), each correlating with retention, as resulted for the analysis of dyes in the Schweppe collection.

A dye which elutes at 15.9 min, with λmax = 410 nm and a molecular weight M = 259 (MS− 258) was present on an orange wool sample but cannot be identified, as these characteristics did not match those of dyes in the database. In three cotton samples, orange and yellow, an inorganic pigment, lead chromate ($PbCrO_4$) was responsible for the colour. In these cases attribution was based on the results of elemental analysis performed by X-ray fluorescence spectroscopy, which revealed the presence of chrome and lead.

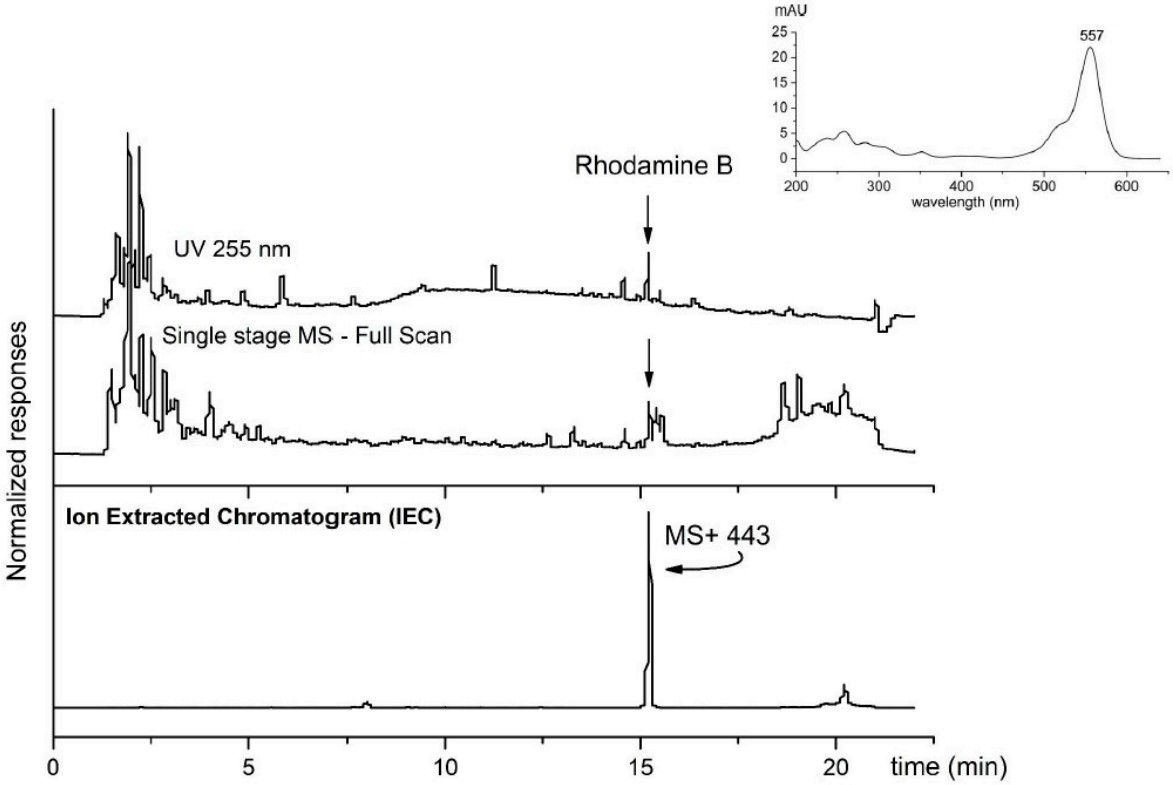

**Figure 2.** Image to support identification of Rhodamine B (1887) by LC-DAD-MS, in shirt inv. 1422 red-pink silk sleeve bracelet decoration (late 19th–early 20th century, Ethnographical Museum, Brasov, Romania).

### 3.3. Green

Apart from the three cases mentioned above, where dyer's broom with a copper mordant or with indigo based dyes were used, synthetic dyes were responsible for the colour. Picric acid (1771), obtained by nitration of phenol and considered one of the first semi-synthetic dyes, was identified in a silk sample which presents a green-yellow hue in a shirt dated 1885 (inv. 3816). As for the other synthetic dyes, detection was based on the UV-Vis spectra (λmax = 360 nm) correlated with retention, according to information acquired by examination of standards. Picric acid was previously identified in a silk sample from a flag used at a celebration in 1871 [50,51]. Brilliant green (1879), a representative of the triarylmethane group, was detected in two samples, on wool and on silk respectively, based on UV-Vis and mass spectrometric data (λmax = 428, 626 nm and M+ 385), associated with retention (Figure 3). According to previous studies, brilliant green was enthusiastically adopted, as proved by its identification in shirts and sheepskin coats from the end of the 19th and beginning of the 20th century [38].

### 3.4. Blue

Twelve samples were described as blue, indigo based dyes being responsible for the colour in more than half (7/12), as well as in the two green samples mentioned above (see Sections 3.2 and 3.3), when they were used together with dyer's broom. Although in most cases (5/7) indigo based dyes were found on wool, they were also used on silk. In all the cases, attribution was based on the detection of indigotin according to UV-Vis spectra (λmax = 238; 285; 330; 610 nm), associated with retention. Indigo based dyes were also the main source of blue in all previous studies dedicated to traditional textiles in Romanian collections [33–38]. Several plants are mentioned in the literature to contain indigotin precursors [52], *Indigofera species* and *Isatis tinctoria* (woad) being the most frequently discussed in relation to European textiles. According to the existing literature [53], with the instrumentation and knowledge available nowadays it is not possible to distinguish

between these species, and neither between natural and synthetic indigo, which became available in 1882. However, as woad was cultivated in Europe for a long time and, under the name *drobșor*, was mentioned in the Romanian literature as dye source at the end of the 19th century, this remains the most likely indigo source.

Synthetic dyes from the triarylmethane group were used in almost half of the total number of samples to achieve a blue hue (5/12), in all but one being used on silk. Victoria Blue B (1883) was responsible for the colour in almost all cases (four/five), on wool and silk while alkali blue (1864) in the remaining silk item. Identification of Victoria Blue B was made based on the UV-Vis and MS data, λmax = 604 nm (MS+ 456), λmax = 616 nm (MS+ 470), correlated with retention, for the chromophores mentioned in the literature (Figure 4) [26]. This dye was also detected on rayon, in the decoration of a Saxon accessory, dated 1935 [36]. The other triarylmethane dye detected, alkali blue, was revealed according to the UV-Vis data, λmax = 596 nm, correlated with retention, as resulted from the analysis of reference dyes in the Schweppe collection.

### 3.5. Brown

Ellagic acid was identified in the two brown silk samples available for analysis, which suggests the use of tannins. Detection of ellagic acid was based on the UV-Vis and mass spectrometric data, λmax = 254, 366 nm, MS− 301. Iron was also present in these samples as revealed by XRF (Figure 5), which is in perfect correlation with the literature which mentions the use of iron salts and tannin-containing plants to dye brown and black. Tannins are polyphenols of vegetal origin, galls on *Quercus species*, *Alnus* barks *and Cotinus coggygria* leaves being among the most popular sources. Recipes to describe the use of tannin sources and iron represent 80% of the formulae for dyeing black and brown in a collection of dyeing recipes edited by the Romanian Academy in 1914 [10]. Alnus bark and green walnut husks and leaves are the most frequent tannin sources used.

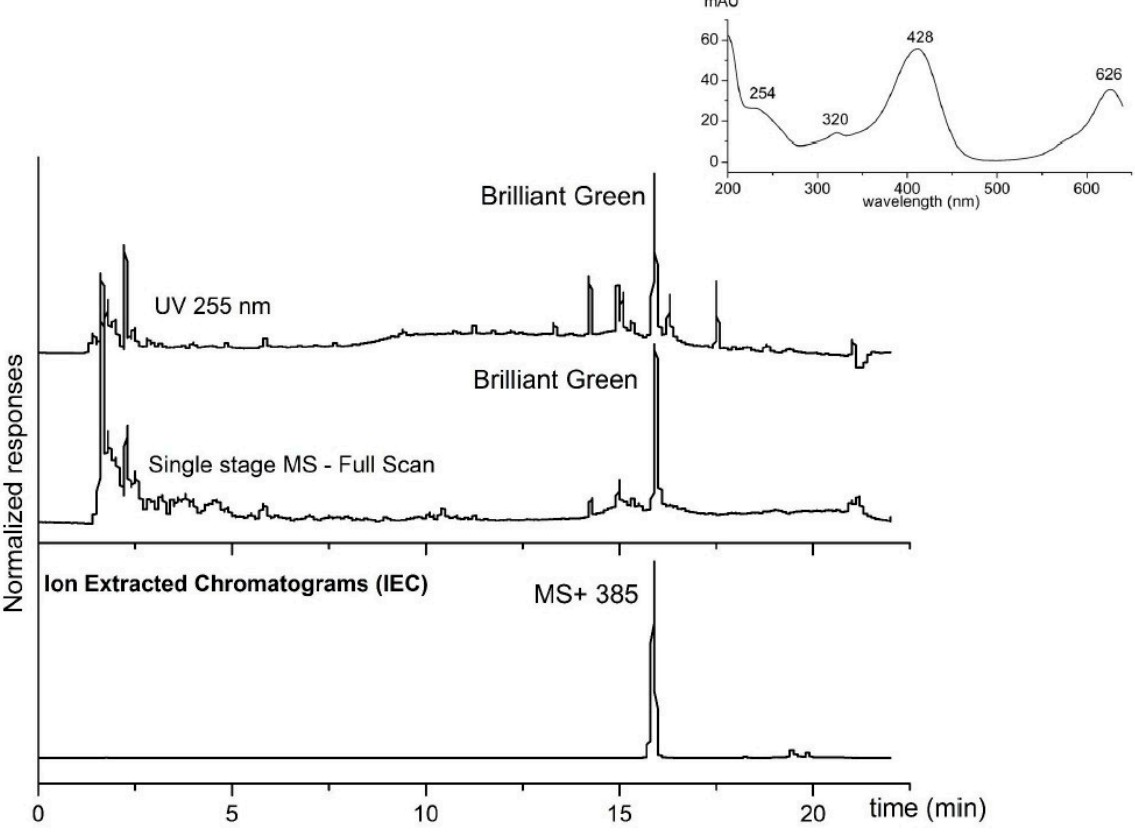

**Figure 3.** Image to support identification of Brilliant Green (1879) by LC-DAD-MS, in shirt inv. 3728 green silk shoulder decoration (last decade 19th–early 20th century, ASTRA Museum, Sibiu, Romania).

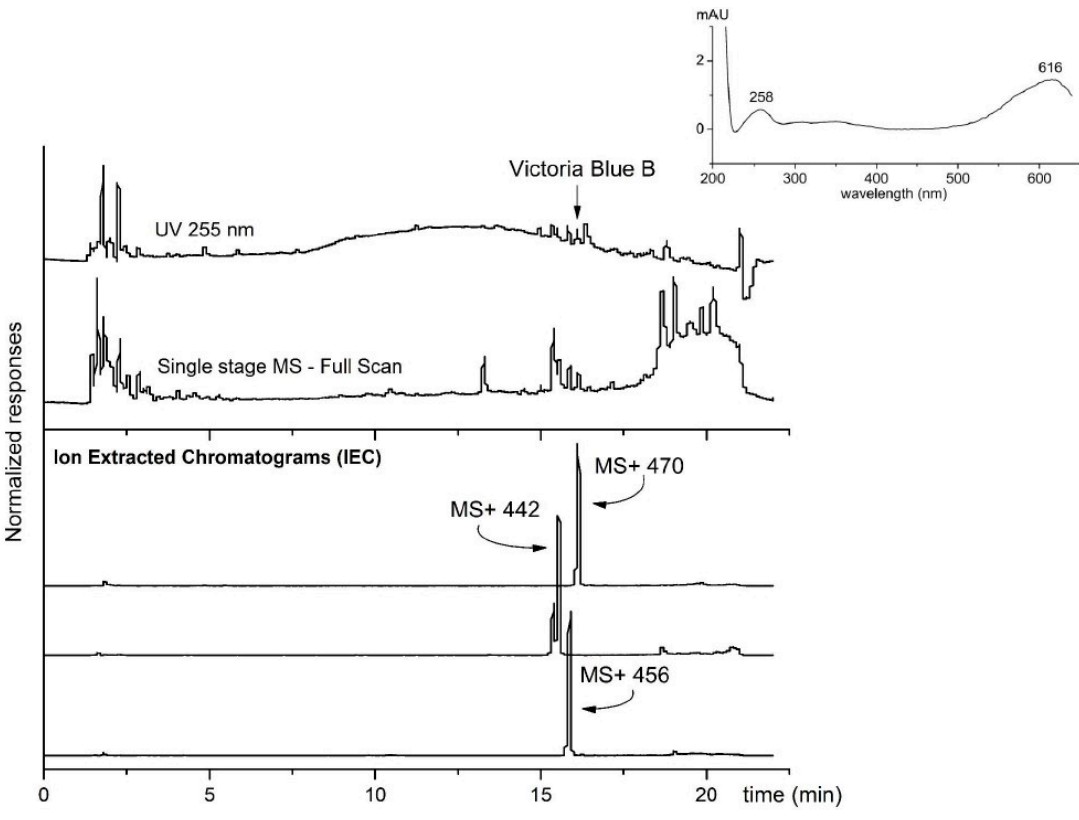

**Figure 4.** Image to support identification of Victoria Blue B (1883) by LC-DAD-MS, in shirt inv. 1429 blue silk shoulder decoration (early 20th century, Ethnographical Museum, Brasov, Romania).

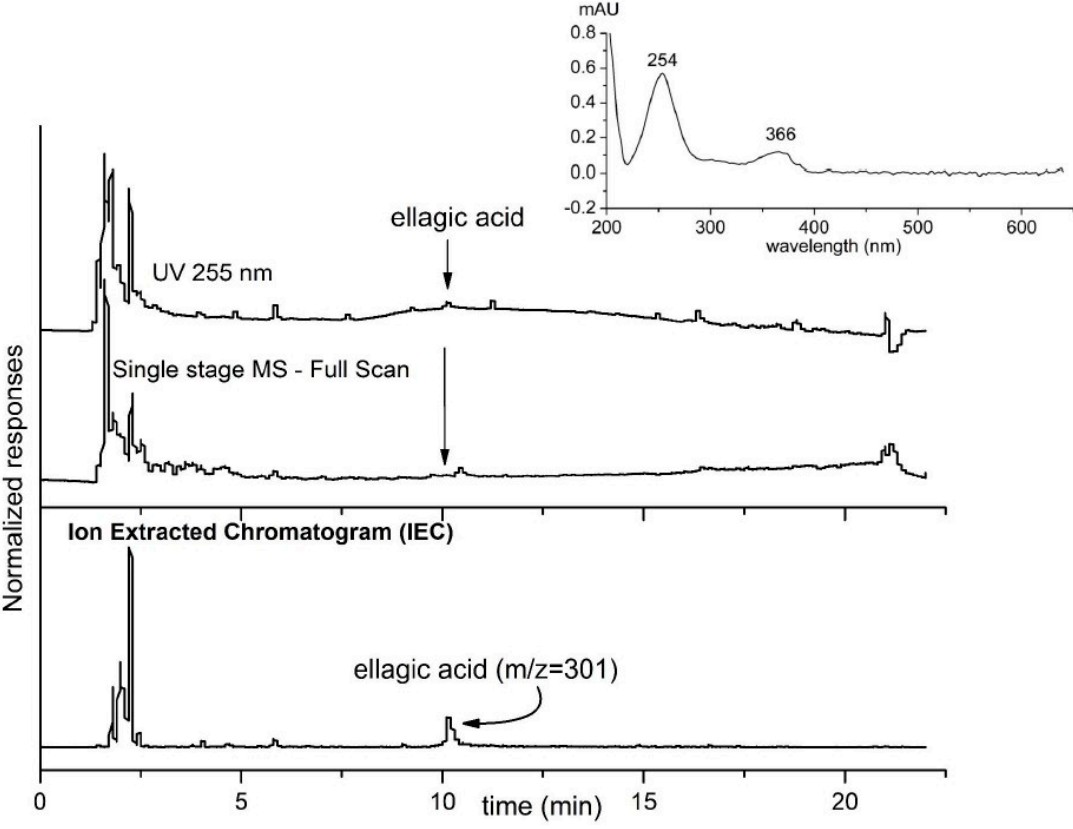

**Figure 5.** Image to support identification of tannins, based on the detection of ellagic acid by LC-DAD-MS, in shirt inv. 194 brown silk shoulder decoration (second half 19th century, ASTRA Museum, Sibiu, Romania).

## 4. Discussion and Conclusions

In total, 50 samples from 13 shirts dated between 1850 and 1930 from the same area in the very center of Romania were studied, in order to monitor transition from natural to the early synthetic dyes. The shirts are hand-made, produced in a small rural area of less than 50 km. A general view of the results, presented for each visual colour of the decoration yarn and correlated to the fiber (Table 4), illustrate the progressive change from natural dyes on wool to synthetic dyes on silk. Although this transition is fast, some phases are worth document, as these observations could be further used in the characterization of undocumented items. The oldest shirts studied, dated in the museum archives as end or late 19th century (inv. 1252, 1415, 6847, 194P, 169P and 20P), are decorated with natural dye sources such as madder, carminic acid based dyes, dyer's broom, sawwort, tannins and indigo dyes. Mineral pigments lead chromate and Prussian blue were added to enlarge the palette of colours with orange or black hues. However, if in the cases of shirts inv. 1252 and 1415, which should be considered the oldest in the study group, dyes are applied on wool (with the exception of lead chromate which is always on cotton), for inv. 169P and inv. 20P natural dyes are used on silk, and should be thus regarded as later. Shirts inv. 194P and inv. 6847, where natural dyes are applied either on wool or on silk, would be placed in between the two groups mentioned. The following step would be represented by shirts decorated with both natural and early synthetic dyes, such as inv. 210 and inv. 1422, applied on wool or on silk. Shirt inv. 3816, which in the museum archives is dated 1885, is worth particular attention, due to its very precise date and the large number of synthetic dyes detected, including two, auramine and Victoria Blue B, introduced (in 1883), two years before the given date. If we consider that, in the absence of written sources, museum dating is usually based on interviews at the moment objects enter the collections, it becomes debatable if 1885 is the correct date for shirt inv. 3816, and whether synthetic dyes (either as dyes or dyed yarns) were available in Transylvanian villages so quickly or, in another scenario, if the date of 1885 should be refined based on literature data regarding the exact moment when each of the dyes detected was traded (not available at the moment of the present study). A last step in the adoption of the early synthetic dyes would be represented by shirts inv. 177P, 3728, 6464 and 1429, dated 19th to 20th century or early 20th century, where a large number of early synthetic dyes are exclusively used, mostly on silk. The use of a natural dye source, carminic acid based dye, in combination with synthetic alizarin in a red silk decoration (shirt inv. 1429) would be seen as an exception, representative of the experiments which characterize the transition period. Apart for documentation and criteria for undocumented objects' characterization, the results obtained represent a valuable resource for the new movement of sewing-groups, interested in reproducing the old shirts as faithfully as possible, including the textile materials and dyes used.

LC-DAD and LC-DAD-MS were used to successfully notify the use of six natural dye sources and to identify 13 early synthetic dyes, many of them for the first time, in textiles from Romanian collections. With reference to the latter detections, their recognition comes to certify the procedure followed and the in-house database used, especially concerning what the analytical experiments collected from samples from the Helmut Schweppe collection. LC-DAD was proved as a useful tool for dye identification in most cases. However, information provided by the mass spectrometer detector, placed in line with the diode array, significantly improves dyes and biological source attribution. These benefits come, on the one hand, from the mass spectrometer's high sensitivity, and on the other from the complementarity of the two detectors. The first attribute will have consequences in better detection of minor compounds and the second provides more identification criteria and confident recognition of the dyes used.

Based on the case study presented, transition from natural to the early synthetic dyes in the second half of the 19th century is for the first time documented, based on analytical arguments. Adoption of the early synthetic dyes was very fast, the high enthusiasm for brighter colours and the larger variety of hues resulting in the complete replacement of the natural sources by the first decades of the 20th century.

**Table 4.** General view on the results obtained to illustrate, for each color, the chronology of the dyes used from the end of the 19th to the early 20th century. For abbreviation, see Tables 1 and 2. The fibers used are given in brackets (C—cotton, W—wool, S—silk). Date attribution is based on the museum archives.

| | Inv. 1252 (End 19) | Inv. 1415 (End 19) | Inv. 194P (Late 19) | Inv. 6847 (Late 19) | Inv. 169P (Late 19) | Inv. 20P (Late 19) | Inv. 210 (Late 19) | Inv. 1422 (19–20) | Inv. 3816 1885 | Inv. 177 (19–20) | Inv. 3728 (19–20) | Inv. 6464 (19–20) | Inv. 1429 (Early 20) |
|---|---|---|---|---|---|---|---|---|---|---|---|---|---|
| red | Mad (W) | Mad (W) | Mad (W) | Mad (W) | ca (S) | ca (S) | ca (S) | Mad (W) / Al (C) (1871) | ca (S) | Me-Vio (W) | Ora GG (1878) (S) | Rhod (1887) + Saf T (1859) (S) | Fuch (1856) (S) / Al (1871) + ca (C) |
| pink/violet | | | ca (S) | | | ca (S) | Me-Vio (1861) (S) | Rhod (1887) (S) | | Rhod (1887) (S) | Fuch Me-Vio (S) | Rhod (1887) (S) | Rhod (1887) (W) |
| orange | PbCrO4 (C) | | | PbCrO4 (C) | | | | | Aur (S) (1883) | | | Uran (S) (1871) | |
| yellow | - | - | | PbCrO4 (C) | | DB + Saw (S) | | DB (W) | Pic Ac (1771) (S) | | | | Azo Fl (1880) (S) |
| blue | Ind (W) | Ind (W) | Ind (W) | Ind (W) | Ind (S) | Ind (S) | | Ind (W) | Vic B (1883) (S) / Alkali Blue (1864) | | Vic B (1883) (S) | Vic B (1883) (S) | Vic B (1883) (S) |
| green | DB Cu (W) | DB Ind (W) | DB Ind (W) | | | lu + Ind (S) | | | Bri Gr (1879) (W) | | Bri Gr (1879) (S) | | |
| brown/black | | | | tan (S) | tan Fe(S) | ca + Pr Blue (S) | | | | | | | |

**Author Contributions:** Conceptualization, I.P.; methodology, I.P. and I.C.T.; sample collection, I.C.T.; investigation, I.P., S.V. and F.A.; writing—original draft preparation, I.P. and I.C.T.; writing—review and editing, I.P., I.C.T., S.V. and F.A. All authors have read and agreed to the published version of the manuscript.

**Funding:** This research received no external funding.

**Data Availability Statement:** Data are available upon request by email to the corresponding author, Irina Petroviciu.

**Acknowledgments:** The authors express their gratitude to Ronnee Barnett, textile restorer at the Metropolitan Museum of New York (MET) who built the Helmut Schweppe collection of early synthetic dyes used as reference database and to Florica Zaharia, Muzeul Textilelor Băița Romania who offered it for research; the ASTRA Museum Sibiu (Ciprian Stefan, Mirela Cretu, Elena Gavan), Ethnographical Museum, Brașov (Alexandru Stanescu, Ligia Fulga, Mihaela Paulic), "Dimitrie Gusti" Village Museum in Bucharest, Bucovina Museum in Suceava and Ethnographical Museum of Transylvania in Cluj for providing access to collections and archives; Agilrom Scientific SRL Romania and IRASM Department in "Horia Hulubei" National Research Institute for Physics and Nuclear Engineering, Romania, who offered acces to the analytical instrumentation and sample preparation facilities; Gheorghe Niculescu for the XRF analysis.

**Conflicts of Interest:** The authors declare no conflict of interest.

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
