# Peer review of "Transition from Natural to Early Synthetic Dyes in the Romanian Traditional Shirts Decoration"

_heritage, doi:10.3390/heritage6010027_

Round 1
Reviewer 1 Report
Materials and Methods:
Lines 153 & 155: CH3OH/H2O NOT CH3OH/H2O
Lines 191/2: same as for System 1, except for Solvent B which was 191 methanol/acetonitrile 1:1 (v/v) with no formic acid content. HOWEVER, Solvent B in system 1 (line 178) has no formic acid either!
Discussion and Conclusion:
Line 411: ‘sensitivity’ NOT ‘sensibility’!
Figures and Tables:
Line 451 and 456: There is no Table 5 (I assume Table 4 is meant).
Line 464: date attribution RATHER THAN data attribution?
Author Response
Dear Reviewer,
Thank you very much for carefully reading our article, I very much appreciate your comments which will further contribute to better valorize our work. We corrected the text according to your observations.
Please find as follows the detailed explanations/ answers to your comments.
All the best wishes,
Irina Petroviciu
Materials and Methods:
Lines 153 & 155: CHOH/HO NOT CH3OH/H2O
Response 1: corrected
Lines 191/2: same as for System 1, except for Solvent B which was 191 methanol/acetonitrile 1:1 (v/v) with no formic acid content. HOWEVER, Solvent B in system 1 (line 178) has no formic acid either!
Response 2: corrected. Sorry, I was wrong, as for System 1, Solvent B was methanol/acetonitrile (1:1, v/v) plus 0.2% formic acid (as solvent B). A did the necessary correction in the text
Discussion and Conclusion:
Line 411: ‘sensitivity’ NOT ‘sensibility’!
Response 3: corrected
Figures and Tables:
Line 451 and 456: There is no Table 5 (I assume Table 4 is meant).
Response 4: corrected
Line 464: date attribution RATHER THAN data attribution?
Response 5: corrected

Reviewer 2 Report
The paper “Transition from natural to early synthetic dyes in the Romanian traditional shirts decoration” accurately characterize dye and textile composition of several traditional shirts of Romanian area providing a full identification of the dye sources used for each coloration. The work is well conceptualized, organized and attentively discussed. The results are interesting, and the aim of the paper successfully aimed. The abstract and introduction are not fluent, I would suggest a careful editing of English language.
In my opinion, the paper is suitable to be published in Heritage after minor revisions.
Suggestions and corrections are following:
Abstract:
I think that abstract can be shortened in order to be more concise and effective. Some information is already in the introduction and can be omitted in the abstract.
· Paragraphs at line 57-60 need to be better clarified. I suggest the following version: “Particularly relevant is the one from Valea Hârtibaciului, an area of Transylvania in the very centre of Romania. Although sober in appearance through the large fields of white plain weave, it is discreetly decorated with elaborated embroideries in the sleeve bracelets, over the shoulders and neck”.
· Line 68-69: please insert “couple with” instead of “with”
· Line 72: substitute “for” with “relative to”
· Line 74: substitute “are” with “were”
Introduction
· Please rephrase line 90-92.
· Paragraphs at line 103-104 need to be better clarified. I suggest the following version: “…from 1740 to 1771, when indigo carmine and picric acid were first synthesized [4]. Nevertheless, the triumph of synthetic dyes started in 1856, when the first synthetic dye, mauveine, was accidentally discovered [4, 5]”.
· Please rephrase line 105-107
· Paragraphs at line 107-109 need to be better clarified. I suggest the following version: “Regarding the period between 1850 and 1900, literature mentions that in the three Romanian provinces (Wallachia, Moldavia and Transylvania) a large number of local plants were employed for dyeing purposes”.
· Line 109: substitute “to document” with “documenting”
· I think that the point of view expressed in line 117-119 is subjective, please rephrase it.
· Please rephrase 121-124.
· Line 136-137: substitute “in the” with “belonging to”.
Materials and methods
· I suggest: 2.1. Sample description, documentation and pretreatment
· Line 103 and 105: use subscript CH3OH/H2O
· Please modify Line 159 as follow: The two solutions were merged and thus analyzed.
Results
Line 244-248: I do not totally agree with this statement. Minority compounds of Rubia tinctorum, such as munjistin, anthragallol, xanthopurpurin and rubiadin, are obviously present in a lower amounts in the sample extract but they are often detected by DAD in ancient textiles [Deyjoo, Roya, et al. "Coptic textiles in Tehran: dye and fibre characterisation in four Coptic textiles preserved at the Moghadam Museum." Archaeological and Anthropological Sciences 13.12 (2021): 1-18], even if its lower sensitivity with respect to MS detectors. Thus, the reasons why they are not detected in the sample can be ascribed both to partial degradation of the sample with aging and due to the acidic hydrolysis (e.g possibly inducing decarboxylation of munjistin [Mouri, Chika, and Richard Laursen. "Identification of anthraquinone markers for distinguishing Rubia species in madder-dyed textiles by HPLC." Microchimica Acta 179.1 (2012): 105-113]) which reduces the amount of the minor components to a level under the detection limit.
Author Response
Dear Reviewer,
Thank you very much for carefully reading our article, I very much appreciate your comments which will further contribute to better valorize our work. We corrected the text according to your observations.
Please find as follows the detailed explanations/ answers to your comments.
All the best wishes,
Irina Petroviciu
Abstract:
I think that abstract can be shortened in order to be more concise and effective. Some information is already in the introduction and can be omitted in the abstract. Paragraphs at line 57-60 need to be better clarified. I suggest the following version: “Particularly relevant is the one from Valea Hârtibaciului, an area of Transylvania in the very centre of Romania. Although sober in appearance through the large fields of white plain weave, it is discreetly decorated with elaborated embroideries in the sleeve bracelets, over the shoulders and neck”.
Response 1: corrected
- Line 68-69: please insert “couple with” instead of “with”
Response 2: corrected
- Line 72: substitute “for” with “relative to”
Response 3: corrected
- Line 74: substitute “are” with “were”
Response 4: corrected
Introduction
- Please rephrase line 90-92.
Response 5: corrected, rephrased
- Paragraphs at line 103-104 need to be better clarified. I suggest the following version: “…from 1740 to 1771, when indigo carmine and picric acid were first synthesized [4]. Nevertheless, the triumph of synthetic dyes started in 1856, when the first synthetic dye, mauveine, was accidentally discovered [4, 5]”.
Response 6: corrected, thank you very much for the suggestion
- Please rephrase line 105-107
Response 7: corrected, thank you very much for the suggestion
- Paragraphs at line 107-109 need to be better clarified. I suggest the following version: “Regarding the period between 1850 and 1900, literature mentions that in the three Romanian provinces (Wallachia, Moldavia and Transylvania) a large number of local plants were employed for dyeing purposes”.
Response 8: corrected, thank you
- Line 109: substitute “to document” with “documenting”
Response 9: corrected
- I think that the point of view expressed in line 117-119 is subjective, please rephrase it.
Response 10: corrected, rephrased
- Please rephrase 121-124.
Response 11: corrected, rephrased
- Line 136-137: substitute “in the” with “belonging to”.
Response 12: corrected
Materials and methods
- I suggest: 2.1. Sample description, documentation and pretreatment
Response 13: corrected, thank you
- Line 103 and 105: use subscript CH3OH/H2O
Response 14: corrected
- Please modify Line 159 as follow: The two solutions were merged and thus analyzed.
Response 15: corrected
Results
Line 244-248: I do not totally agree with this statement. Minority compounds of Rubia tinctorum, such as munjistin, anthragallol, xanthopurpurin and rubiadin, are obviously present in a lower amounts in the sample extract but they are often detected by DAD in ancient textiles [Deyjoo, Roya, et al. "Coptic textiles in Tehran: dye and fibre characterisation in four Coptic textiles preserved at the Moghadam Museum." Archaeological and Anthropological Sciences 13.12 (2021): 1-18], even if its lower sensitivity with respect to MS detectors. Thus, the reasons why they are not detected in the sample can be ascribed both to partial degradation of the sample with aging and due to the acidic hydrolysis (e.g possibly inducing decarboxylation of munjistin [Mouri, Chika, and Richard Laursen. "Identification of anthraquinone markers for distinguishing Rubia species in madder-dyed textiles by HPLC." Microchimica Acta 179.1 (2012): 105-113]) which reduces the amount of the minor components to a level under the detection limit
Response 16: corrected, rephrased, thank you
